# Ego or Eco? Neither Ecological nor Egoistic Appeals of Persuasive Climate Change Messages Impacted Pro-Environmental Behavior

**Jana Sophie Kesenheimer \*** and **Tobias Greitemeyer**

Institute for Psychology, University of Innsbruck, Innrain 52, 6020 Innsbruck, Austria;
tobias.greitemeyer@uibk.ac.at

**\*** Correspondence: jana.kesenheimer@uibk.ac.at

**Abstract:** Based on the 'Inclusion Model of Environmental Concern', we tested whether daily messaging intervention increases participants' pro-environmental behavior (PEB). In a two (time: pre vs. post, repeated measure) × three (condition: egoistic appeals, ecological appeals, control group) experimental design, two hundred and eighteen individuals received either daily messages containing egoistic appeals for action to prevent climate change (e.g., preventing personal consequences of released diseases in melting arctic ice), ecological appeals (e.g., ecological consequences of melting glaciers), or no messages (control). PEB was assessed via self-reports and donations to an environmental organization. Neither of the appeals had an effect on the two dependent measures. Irrespective of experimental conditions, self-reported PEB was higher in the post- compared with the pre-test. Overall, the present results do not provide support for the effectiveness of a daily messaging technique. Instead, it appears that 'being observed' is the more effective 'intervention'. Implications for how to foster PEB are discussed.

**Keywords:** pro-environmental behavior; intervention; persuasion

## 1. Introduction

As climate change is human-caused [1], the solution of global climate crisis lies in human hands. Pro-environmental behavior (PEB) means minimizing environmental impact or benefitting the environment through human behavior and is increasingly becoming a vital factor in environmental psychology. Interventions to promote PEB have previously focused on informational strategies to increase awareness, shape attitudes, and, as a result, change human behavior [2]. By definition, informational strategies for promoting PEB do not actually change external contexts but focus on feedback, goal setting, and persuasion principles, among others [2,3]. Persuasion means providing information with the aim to change attitudes or behavior. However, persuasion has been predominately used as a one-time intervention by presenting video or word stimuli. To achieve higher message effectiveness, the present study employed daily messaging intervention during a five-day period.

We are all targets of a daily flood of messages. Among the information reaching us in daily life through all sorts of media and adverts, there are numerous messages relating to climate crisis and sustainability (e.g., [4]). Therefore, it is important to investigate *which* information, and *if* such information, can trigger PEB in daily life. In previous research, persuasive interventions typically focused on PEB appeals, consisting of, for example, monetary vs. moral [5], self-interest vs. altruism [6], or egoistic vs. biospheric motives [7]. These attempts reflected values of self-enhancement on the one hand and self-transcendence on the other, according to a bipolar value spectrum [8]. In the context of PEB, this duality has been addressed in the 'Inclusion Model of Environmental Concern' [9].

The model differentiates between biospheric and egoistic needs that can be satisfied through PEB. It has been claimed that egoistic appeals are superior in their effectiveness as they are of common interest, whereas biospheric needs affect only people with a strong biospheric value orientation.

We adapted the term of biospheric appeals to the term 'ecological', as we focused on impacts on the natural environment. As an example, an ecological appeal of drinking tap water is to avoid plastic and reduce carbon emissions that are released in the production and transportation process of bottled water. Conversely, we use the terminology 'egoistic' for appeals that primarily address a person's own welfare. Egoistic aspects can also motivate behavior to reduce carbon dioxide emissions, but this occurs independently of any environmental reasons. For example, one can prefer drinking tap water in order to save money. Given that both egoistic and ecological motives contribute to people's PEB, we anticipated that participants that received either type of daily messages would be more likely than participants in a control group receiving no intervention to increase their PEB.

## 1.1. Previous Research on the Use of Persuasive Messages to Promote PEB

The effectiveness of daily messaging interventions has been successfully documented in previous research. In one study [10], the authors sent out fourteen daily messages concerning the health and/or environmental benefits of reducing red meat consumption. The health condition and the environmental condition (but not health + environment) were successful in shaping participants' attitudes, which then led to a reduction of red meat consumption. Participants of a control group that were not receiving an intervention showed no changes in behavior. The same methodology was also successful in reducing red meat consumption by the use of emotional (but not informational) daily messages [11]. Similarly, we assumed that the daily messaging technique could successfully increase PEB. Although there has been growing interest in promoting PEB, the effectiveness of a daily messaging intervention has not yet been investigated.

In previous research, there has been disagreement regarding the contents of persuasive messages that enhance PEB. De Dominics and colleagues [6] highlighted the role of self-interest vs. altruism as individual motives in shaping PEB. As a dependent variable, the authors assessed the intention to conserve energy and use public transit. Three studies provided evidence that personal benefits appealed to all people, whereas environmental benefits only influenced altruistic people. The finding that the effectiveness of environmental appeals depended on people's level of altruism is in line with the 'Inclusion Model of Environmental Concern', which suggests that satisfying egoistic needs is of common interest, but biospheric needs are limited to people with strong biospheric value orientations. However, other researchers [5] reasoned that biospheric values were most successful in promoting PEB (in their study, they checked tire pressure for environmental reasons), because people aim to maintain a favorable view of themselves. In contrast, results of another study [7] showed that egoistic appeals worked equally well for biospheric and egoistic people (in minimalizing living). Biospheric appeals always yielded a null effect. Given that previous findings led to mixed results regarding the effectiveness of persuasive appeals to promote PEB, we tested both egoistic and ecological messages.

## 1.2. Potential Moderator Variables

On an exploratory basis, we examined whether some people would be positively affected by receiving egoistic or ecological appeals more than others. A first moderating variable could be participants' social value orientation. Research has shown that people differ in their preferences to enhance equality in outcomes (prosocial orientation), or seek to enhance their own outcomes in absolute terms (individualistic/proself orientation) or comparative terms (competitive orientation) [12]. Social value orientations are typically assessed in decomposed games, where one has to decide outcomes for one's self and for another (fictive) person. Previous research [13] has shown that social value orientations, proself vs. prosocial in particular, are related to a person's PEB. Prosocials, compared to proselfs, were highly aware of the social-altruistic consequences of PEB but less aware of the egoistic consequences [13]. Indeed, as already outlined, De Dominic et al. [6] showed that a fit of values and

the presented appeals of PEB (personal vs. environmental benefits) had a positive impact on behavior change. It should be noted, however, that in another study [7] a fit of social values and persuasive messages did not affect the intervention's effectiveness. We thus formulated the potential moderating impact of the fit between social value orientations and persuasive messages to promote PEB only as a research question (instead of a hypothesis).

Further, a pro-environmental attitude is one of the most important positive predictors of PEB [14]. We reasoned that people with a strong pro-environmental attitude could be especially sensitive to ecological appeals, and thus they would be particularly positively affected by receiving them. In contrast, egoistic appeals should have no beneficial effects for participants with a strong pro-environmental attitude. Similarly, previous research showed that strong consideration of future consequences when accepting and responding to anthropogenic climate change enhances commitment to acting pro-environmentally [15]. Time perspective (consideration of distant vs. immediate consequences) and, consequently, the acceptance of human-made global climate change have been shown to influence environmental mitigation behavior. Thus, we also examined whether consideration of future consequences would positively influence the effectiveness of ecological appeals. As with regard to the impact of social value orientations, we formulated research questions (rather than hypotheses) because of possible ceiling effects that diminish the effectiveness of the persuasive appeals. That is, individuals with a strong pro-environmental attitude and those who strongly consider the potential future outcomes of their current behavior already behave in an environmentally friendly way, which leaves little room for the persuasive appeals to further promote their PEB.

### 1.3. The Present Study

The present investigation uses the technique of daily messaging with the aim of promoting PEB. Concretely, we investigated the influence of five daily messages containing either egoistic or ecological appeals for PEB. We compared the effectiveness of these appeals with a control group where participants did not receive any messages. A fit between social value orientation and persuasive messages, pro-environmental attitude, and consideration of future consequences were considered as potential moderator variables. We pre-registered (https://aspredicted.org/iv8ab.pdf; before any analyses were performed, we calculated another power analysis as the expected effect size was lowered from $f = 0.23$ to $f = 0.20$ and thus we recruited more participants.) the following main hypotheses:

**Hypotheses 1 (H1).** *Participants receiving interventional messages show stronger improvement of their self-reported PEB (pre to post) compared with a control group that is receiving no intervention.*

**Hypotheses 2 (H2).** *Participants receiving interventional messages are more likely to donate to an environmental organization compared with a control group that is receiving no intervention.*

We further tested the following research questions (also pre-registered):

RQ1: Does a fit of social value orientation (proself vs. prosocial) and content (egoistic vs. ecological appeals) strengthen the influence of interventional messages on PEB? A fit means a proself value orientation in combination with egoistic appeals on the one hand and a prosocial value orientation in combination with ecological appeals on the other.

RQ2: Does a strongly endorsing pro-environmental attitude strengthen the influence of ecological appeals on PEB?

RQ3: Does strong consideration of future consequences strengthen the influence of ecological appeals on PEB?

## 2. Methods

### 2.1. Participants

Participants were recruited online by adverts in newsletters (University Newsletter, "Initiative of Psychology in Environmental Protection" and "Psychology Today" magazine) and social media forums (Facebook). The invitation stated that it was a study about "personality and behavior in everyday life". To minimize selection biases, the invitation did not say that the study was about environmental attitudes or behavior. Then, participants were randomly assigned to one of the three experimental conditions. During the following five days, participants in the experimental groups received messages on their smartphones. Afterwards, all participants received an e-mail to take part in the second part of the study, which assessed their PEB for the last five days (the study "period"). In this second part (post), participants of the experimental groups also responded to an attention check. They were asked if specific contents (e.g., melting of arctic ice [true], electrified motorization [false]) were part of the messages they received.

A G*Power Analysis [16] (MANOVA for a two times repeated measurement in three groups including interaction effects, 95% Power) revealed a sample size of 198 participants for an expected medium effect size of $f = 0.20$, based on previous findings from Carfora et al. [10,11] who found that both health and environment daily messaging conditions reduced red and processed meat consumption compared with a control condition ($d = 0.53$, $d = 0.47$, respectively). We slightly oversampled because of possible dropouts. Initially, three hundred and fifty-three participants took part during the first measure of the study. With a dropout rate of 38.25%, two hundred and eighteen of them (152 female, 64 male and three people with diverse gender) participated in both parts of the study. In line with the pre-registered exclusion criteria, twelve participants were excluded because they failed attention checks. They failed to correctly answer more than two (out of seven) questions concerning the contents of the persuasive messages. The resulting sample consisted of 147 women, 57 men and two diverse gender. There were 63 participants that received egoistic appeals, 61 participants received egocentric appeals, and 82 participants were in the control group. The age of participants ranged from 18 to 64 years ($M = 26.03$, $SD = 8.05$). Most of them were highly educated, as 44.66% ($n = 92$) had a university degree and another 32.04% held a high school degree. More than half of the participants lived in Austria ($n = 106$), 83 lived in Germany, and 17 lived in other European countries. Mean political orientation was 4.06 ($SD = 1.63$), presented on a slider from "1-politically left" to "11–politically right".

### 2.2. Design

The present study had a two (time: pre vs. post) × three (condition: egoistic appeals, ecological appeals, control group) experimental design with repeated measures on the time factor. The experimental conditions differed by the contents of messages that participants received. Messages included egoistic or ecological appeals for PEB. A control group received no persuasive messages.

### 2.3. Materials

Participants in the experimental groups received one message per day via Telegram Channel (© 2020 Telegram—messenger for iPhone, Android and Windows). Each message was sent out automatically by a Chatbot at 8 a.m. In a pilot study, we validated our messages. First, we composed 18 contents that contained either ecological or egoistic appeals (nine each) for climate change prevention. Among these were the melting of glaciers, the use of resources in daily life (water, energy), and the carbon dioxide proportion in the air. Twenty-eight participants (21 female, 6 male, one person with diverse gender; $M_{age} = 32.04$, $SD_{age} = 12.73$) indicated to what extent each of the 18 messages relates to ecological or egoistic appeals ("This message is of ecological manner/relates to our collective environment"; "This message is of egoistic manner/relates to my individual life"). Answers were given on a seven-point-scale (1—"not at all agree", 7—"fully agree"). In the main study, we used the ten

messages (five egoistic, five ecological) that were most clearly of either an egoistic or ecological manner. These messages and statistical test results are reported in Appendix A (original materials in German language can be found online: https://osf.io/xkcf3).

To assess participants' PEB, we relied on two measures. The self-reported PEB of participants was assessed by a questionnaire before (pre) and after (post) the intervention (all items and information about reliability can be found in Appendix B). The questionnaire was based on previously validated scales [17,18] and some newly developed items for this study. All items targeted behaviors that are undertaken primarily for environmental reasons (Appendix B). Self-report measures of PEB are known to have validity issues [19]. Indeed, people tend to overestimate their PEB because of social desirability effects [20]. Hence, we also employed an objective measure of the participants' PEB. Participants learned that they could take part in a raffle to win up to 50 Euros. They were asked whether they would like to donate part (or all) of this money to a pro-environmental organization (if chosen as a winner, the researchers would donate the money to the organization). This measure was only assessed in the post-test. The participants also had the choice to donate the prize money to a prosocial organization. We assessed the participants' willingness to donate parts of the potential prize money to prosocial organizations ("NEUstart", the Austrian probation service or "Austrian Cancer Aid") and/or pro-environmental organizations ("Protect our Winters Austria" or "Fridays for Future Austria"). We anticipated that participants that received either type of daily messages would be more likely than participants in the control group to donate to a pro-environmental organization. That is, the interventional messages were assumed to have a specific effect on the participants' PEB but not to influence their general propensity to benefit others [21].

As potential moderator variables, we assessed the participants' social value orientation [12], pro-environmental attitude (New Ecological Paradigm scale, NEP, [22]), and consideration of future consequences [18]. A description of scales including sample items is given in Table 1. These variables were all assessed at time one (before participants received any messages). Regarding the participants' social value orientation, the decomposed game [12] included nine decisions about a distribution of points to either oneself or another person. The predominant type of decision was coded as the individual's social value orientation. In total, there were 37 participants with a proself orientation, 157 with a prosocial orientation, 8 with a competitive orientation and 16 participants with an inconclusive orientation.

**Table 1.** Scales sample items, means, standard deviations, and reliability information. Items were presented in the German language.

| Name | Scale | Sample Item | Mean | SD | Cronbach's $\alpha$ [CI95%] |
|---|---|---|---|---|---|
| NEP [22] | 1–5 (don't agree at all–fully agree) | *Humans are born to rule over nature.* | 3.93 | 0.41 | 0.69 [0.63, 0.75] |
| Consideration of distant future consequences [18] | 1–5 (don't agree at all–fully agree) | *Generally, my behavior is influenced by future consequences.* | 3.81 | 0.49 | 0.57 [0.48, 0.65] |
| Consideration of immediate consequences [18] | 1–5 (don't agree at al –fully agree) | *I only act to satisfy immediate concerns, figuring the future will take care of itself.* | 2.00 | 0.63 | 0.65 [0.58, 0.72] |
| Self-reported PEB pre | 1–7 (never–always) | *In the last 5 days, I have payed attention to labels for ecological products (e.g., on food, clothes).* | 5.86 | 1.14 | 0.77 [0.72, 0.81] |

**Table 1.** *Cont.*

| Name | Scale | Sample Item | Mean | SD | Cronbach's α [CI95%] |
|---|---|---|---|---|---|
| Self-reported PEB post | 1–7 (never–always) | *In the last 5 days, I have payed attention to labels for ecological products (e.g., on food, clothes).* | 6.00 | 1.11 | 0.80 [0.76, 0.84] |

*Note.* Pro-Environmental Behavior (PEB); 95% confidence intervals [CI95%]; scales on PEB are fully attached in the appendix.

### 2.4. Data Analysis

We coded fit by "1" ($n = 63$) when participants with a primarily prosocial value orientation received ecological appeals and when participants with a primary individualistic value orientation received egoistic appeals, respectively. Non-fit was coded with "0" ($n = 61$). To investigate RS2 and RSQ 3, data of participants that received ecological messages were coded with "1", otherwise with "0".

## 3. Results

### 3.1. Descriptive Analyses

Normal distribution was violated for each scale other than NEP, as Shapiro-Wilk tests showed. Therefore, non-parametric raw correlations were conducted, which are reported in Table 2. Figures 1 and 2 show means and standard errors for the dependent variables for each of the experimental groups. Groups did not differ in terms of age (F (2, 203) = 1.00, $p = 0.370$, $\eta^2 = 0.01$), gender ((F (2, 203) = 0.33, $p = 0.721$, $\eta^2 = 0.00$), education (F (2, 203) = 0.65, $p = 0.526$, $\eta^2 = 0.01$), political attitude (F (2, 203) = 2.28, $p = 0.105$, $\eta^2 = 0.02$), and pro-environmental attitude (NEP mean; F (2, 203) = 0.81, $p = 0.445$, $\eta^2 = 0.01$). In contrast, the experimental groups differed in PEB at time one, F (2, 203) = 3.39, $p = 0.036$, $\eta^2 = 0.03$. At Time 1, the control group reported more frequent PEB than the ego group ($p = 0.034$) and the eco group ($p = 0.024$). At time two, this difference disappeared, F (2, 203) = 1.92, $p = 0.150$, $\eta^2 = 0.02$.

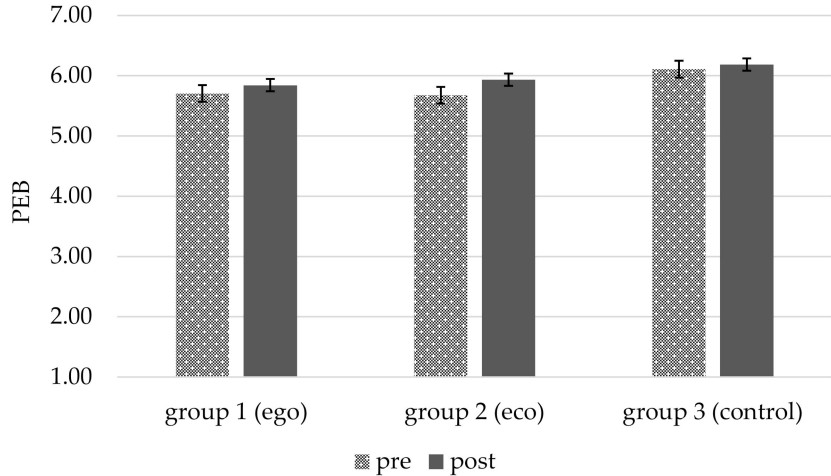

**Figure 1.** Means and standard errors (95%) of the participants' self-reported PEB (pre vs. post).

Table 2. Spearman raw correlations between variables.

| | Age | Gender | Political Attitude | NEP | Cons. of Future Cons. | Cons. of Immediate Cons. | Self-Reported PEB Pre | Self-Reported PEP Post | Willingness to Donate (Prosocial) |
|---|---|---|---|---|---|---|---|---|---|
| Gender | 0.13 [−0.00, 0.26] | | | | | | | | |
| Political attitude | 0.02 [−0.11, 0.16] | 0.04 [−0.09, 0.17] | | | | | | | |
| NEP (pro-environmental attitude) | −0.06 [−0.19, 0.08] | −0.17 * [−0.31, −0.03] | −0.22 ** [−0.35, −0.08] | | | | | | |
| Cons. of future cons. | −0.05 [−0.18, 0.09] | −0.10 [−0.24, 0.05] | −0.10 [−0.25, 0.03] | 0.21 ** [0.07, 0.34] | | | | | |
| Cons. of immediate cons. | 0.03 [−0.12, 0.17] | 0.14 * [0.01, 0.27] | 0.22 ** [0.08, 0.35] | −0.25 *** [−0.38, −0.11] | −0.54 *** [−0.64, −0.43] | | | | |
| Self-reported PEB pre | 0.05 [−0.09, 0.19] | −0.12 [−0.25, 0.01] | −0.21 ** [−0.35, −0.06] | 0.11 [−0.03, 0.25] | 0.47 *** [0.36, 0.56] | −0.23 *** [−0.38, −0.08] | | | |
| Self-reported PEB post | 0.01 [−0.13, 0.15] | −0.17 * [−0.30, −0.04] | −0.20 ** [−0.33, −0.07] | 0.18 ** [0.03, 0.31] | 0.44 *** [0.32, 0.54] | −0.27 *** [−0.40, −0.14] | 0.84 *** [0.78, 0.88] | | |
| Willingness to donate (prosocial) | −0.06 [−0.20, 0.07] | −.09 [−0.22, 0.05] | −0.00 [−0.15, 0.13] | −0.13 [−0.27, 0.00] | −0.09 [−0.23, 0.05] | 0.04 [−0.11, 0.18] | 0.13 [−0.00, 0.25] | 0.08 [−0.06, 0.22] | |
| Willingness to donate (pro-environmental) | 0.09 [−0.05, 0.22] | −0.22 ** [−0.34, −0.09] | −0.10 [−0.23, 0.02] | 0.14 * [0.01, 0.29] | 0.23 *** [0.10, 0.35] | −0.21 ** [−0.34, −0.07] | 0.33 *** [0.20, 0.44] | 0.29 *** [0.16, 0.42] | 0.31 *** [0.17, 0.44] |

*Note.* Pro-Environmental Behavior (PEB); consideration of consequences (Cons. of cons.). Brackets indicate 95% confidence intervals. Level of significance: * $p < 0.05$, ** $p < 0.01$, *** $p < 0.001$.

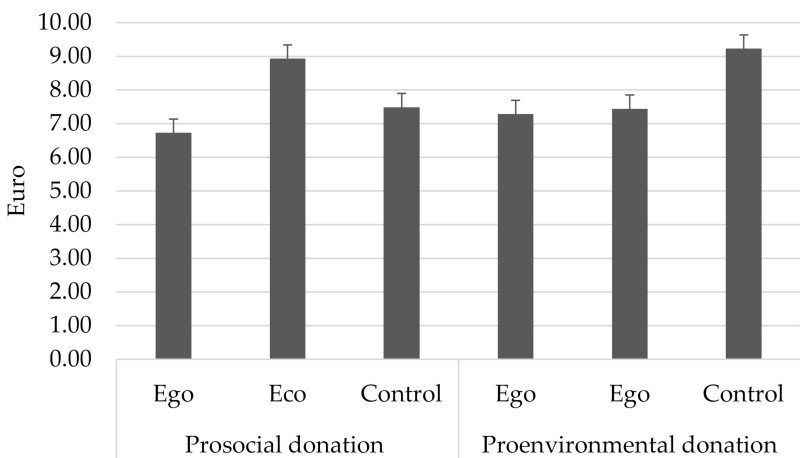

**Figure 2.** Means and standard errors (95%) of the participants' willingness to donate.

### 3.2. Analysis of the Main Hypotheses

To test whether interventional messages enhance PEB, a two (time: pre vs. post) × three (condition: egoistic appeals, ecological appeals, control group) factorial analysis of variance with repeated measures on the first factor was conducted, with self-reported PEB as a dependent variable. Results showed a significant main effect of time, $F(1, 203) = 12.17$, $p = 0.001$, $\eta^2 = 0.06$. At time two (post), PEB was higher (M = 5.99, SE = 0.08) than at the beginning of the study (pre; M = 5.83, SE = 0.08). Neither the main effect of the groups ($F(2, 203) = 2.78$, $p = 0.064$, $\eta^2 = 0.03$) nor the interaction between time and group was significant ($F(2, 203) = 1.37$, $p = 0.250$, $\eta^2 = 0.01$).

However, when exploratorily comparing the simple main effect of time in each group, only the group receiving ecological messages showed a significant change of PEB between measures (pre–post; $M_{pre}$ = 5.68, SD = 1.18, $M_{post}$ = 5.93, SD = 1.09; $t(59) = 3.14$, $p = 0.003$, r = 0.38). There was also an increase in PEB in the other two groups, but it was neither significant in the group receiving egoistic messages ($M_{pre}$ = 5.71, SD = 1.13, $M_{post}$ = 5.84, SD = 1.17; $t(61) = 1.69$, $p = 0.10$, r = 0.21) nor in the control condition ($M_{pre}$ = 6.11, SD = 1.09, $M_{post}$ = 6.19, SD = 1.06; $t(80) = 1.09$, $p = 0.278$, r = 0.12).

Regarding the participants' willingness to donate to a pro-environmental organization, an analysis of variance did not show any differences between groups (ego vs. eco vs. control), $F(2, 203) = 0.61$, $p = 0.547$, $\eta^2 = 0.01$. Likewise, the willingness to donate to a prosocial organization did not differ between the conditions, $F(2, 203) = 0.48$, $p = 0.617$, $\eta^2 = 0.01$. Overall, neither H1 nor H2 were supported by the data.

### 3.3. Analysis of the Research Questions

According to our pre-registration, we had three further research questions. In RQ1, we tested whether a fit of social-value orientation and the content of the persuasive messages would alter the change in PEB (pre vs. post intervention). However, in a two (time) × three (condition) × two (fit) factorial analysis of variance with repeated measures on the first factor, neither the main effect of 'fit', $F(1, 201) = 1.67$, $p = 0.198$, $\eta^2 = 0.01$, the interaction between time (pre vs. post) and fit (fit vs. non fit), $F(1, 201) = 0.00$, $p = 0.974$, $\eta^2 = 0.00$, nor the interaction between time, fit and group (ego vs. eco vs. control), $F(1, 201) = 0.44$, $p = 0.508$, $\eta^2 = 0.00$, was significant. Likewise, fit of social value orientation did not moderate the impact of experimental conditions on pro-environmental donations, b = 4.22, $t(2, 198) = 1.20$, $p = 0.231$. The main effect of fit was also not significant, b = −10.65, $t(2, 198) = 1.62$, $p = 0.106$.

On an exploratory basis, further analyses showed that proself participants reported significantly less PEB (M = 5.68, SD = 1.28) at the post measure than prosocial participants (M = 6.12, SD = 1.01; $p = 0.025$). At the first time of measurement (pre), in contrast, there were no significant

differences ($M_{proself}$ = 5.53, $SD_{proself}$ = 1.25; $M_{prosocial}$ = 5.92, $SD_{prosocial}$ = 1.10; $p$ = 0.342). Accordingly, pro-environmental donations did not significantly differ between proself participants (M = 5.76, SD = 11.50) and prosocial participants (M = 8.99, SD = 12.25; $p$ = 0.145).

To investigate RQ2 and RQ3, participants' PEB difference (post–pre) was calculated. We first examined whether participants' level of pro-environmental attitude would moderate the influence of ecological persuasive messages on PEB (RQ2). A multiple linear regression model revealed no significant main effect of pro-environmental attitude on behavior change, b = 0.63, t (2, 202) = 1.50, $p$ = 0.134. Additionally, no significant interaction between pro-environmental attitude and experimental groups (eco vs. non-eco) was found, b = −0.23, t (2, 202) = −0.98, $p$ = 0.329. However, whereas the relationship between pro-environmental attitude and PEB was significant for the participants who received ecological messages, b = 0.40, t (59) = 2.03, $p$ = 0.047, this relationship was not significant for the participants who received egoistic appeals (b = 0.06, t (61) = 0.06, $p$ = 0.751) or the participants in the control group (b = 0.28, t(80) = 1.57, $p$ = 0.121). With regard to the objective PEB measure, there was no significant interaction between pro-environmental attitude and experimental condition, b = −0.96, t (2, 200) = 0.97, $p$ = 0.691. The main effect of pro-environmental attitude was also not significant, b = 4.77, t (2, 200) = 0.90, $p$ = 0.371.

Lastly, we examined whether consideration of future consequences would moderate the influence of ecological persuasive messages on PEB change. A linear multiple regression showed no significant main effect of consideration of future consequences on behavior change (b = 0.08, t (2, 202) = 0.19, $p$ = 0.852). The interaction with experimental conditions (eco vs. non-eco) was also not significant (b = −0.06, t (2, 202) = −0.27, $p$ = 0.479). Also, there was no significant relation between consideration of future consequences and behavior change in any of the experimental groups (ego: b = 0.08, t (61) = 0.46, $p$ = 0.647; eco: b = 0.02, t (59) = 0.09, $p$ = 0.932; control: b = −0.09, t (80) = −0.71, $p$ = 0.482). Finally, consideration of future consequences did not moderate the impact of experimental condition willingness to donate to pro-environmental organizations, b = −1.49, t (2, 200) = 0.74, $p$ = 0.461. The main effect was also not significant, b = 9.05, t (2, 200) = 1.86, $p$ = 0.064.

## 4. Discussion

In the present study, we tested whether a five-day intervention of daily persuasive messages would promote participants' PEB. Overall, there was a significant effect of time (pre vs. post), showing a general increase of the participants' self-reported PEB. Contradicting our main hypothesis, however, this increase was independent of group assignment. It thus appears that the main finding of our study is that study participation itself enhanced PEB. Such research participation effects [23] have been coined the "question-behavior effect" [24,25], describing the pattern where participants change their behavior when questions are being asked about it. A mechanism for these research participation effects is that the awareness of being observed and that one's own behavior is being judged shapes behavior in a socially accepted manner [26] and could likewise explain why participants in all our experimental groups reported having increased their PEB. Likewise, regarding the objective measure of the participants' PEB (H2), there were no significant differences between the experimental conditions. Participants in the two experimental conditions were not more willing to donate to an environmental organization than the participants in the control condition. It is reassuring that the participants' self-reported PEB and the objective measure of their willingness to donate to an environmental organization were significantly correlated. Overall, however, the present study does not confirm the 'Inclusion Model of Environmental Concern' [9], which suggests that satisfying egoistic needs is of common interest and thus egoistical appeals should be more effective in promoting PEB than ecological appeals.

In terms of potential moderating variables, a fit of social value orientation and persuasive message type did not influence behavior change. Contrary to our research question 1, results did not show that a proself value orientation in combination with egoistic appeals on the one hand or a prosocial value orientation in combination with ecological appeals on the other are especially effective in changing behavior. Prosocial participants reported more PEB than proself participants at time 2 but not at time

1. However, this change was irrespective of the participants' experimental condition. With regard to research question 2, participants' pro-environmental attitude did not affect to what extent they were influenced by receiving ecological messages. When receiving ecological messages, there was a positive relationship between pro-environmental attitude and PEB. For the other two conditions, this relationship was not significant. However, given that the interaction between pro-environmental attitude and experimental group was not significant, it cannot be concluded that the relationship between pro-environmental attitude and PEB was stronger in the ecological messages condition. Lastly, even though consideration of future consequences was highly correlated with PEB, it did not positively affect behavior change, as was the reasoning of research question 3. It strongly correlated with PEB at each point of measure but was not related to stronger increases in PEB. With regard to the objective measure of participants' PEB, neither fit of social value orientation, pro-environmental attitude, nor consideration of future consequences moderated the impact of the experimental condition. Overall, none of the research questions received support from the data.

*4.1. Implications*

Our findings did not provide supportive data for the 'Inclusion Model of Environmental Concern'. Previous research has found inconclusive evidence [5–7]. To be sure, the present study had some limitations (see below), which should be kept in mind when drawing conclusions about the validity of the model. Importantly, however, the present data did not show that egoistical messages would be more effective in promoting PEB than ecological messages. If anything, the trend was in the opposite direction. Future research addressing potential moderating factors that explains under what circumstances egoistical messages are particularly effective (and under what circumstances they are not) would be welcome.

With regard to practical implications, our findings failed to show that receiving persuasive messages would promote PEB. Rather, our findings suggest that research participation itself is a stronger 'intervention'. Therefore, the present study contributes to previous heterogeneous findings on observation effects on PEB. For example, Lange, Brick, and Dewitte [26] did not find support "for an effect of observability on environmental conservation in the laboratory" (p. 1). In contrast, other studies did find that people's PEB is affected by the mere awareness of being observed. For example, images of "watching eyes" that introduced observability led to reduced littering on a university campus when a large number of people were around [27]. As regarding the effectiveness of egoistical messages, future work is needed to clarify under what circumstances observation effects of PEB occur and under which circumstances they are less likely.

*4.2. Limitations and Future Research*

This study has some limitations. First, the sample primary consisted of young and highly educated women, which could explain why values of PEB were already high from the beginning, as it has been shown that women behave in an environmentally friendly way more than men [28] and that education has a positive impact on PEB [29]. This ceiling effect, in turn, may account for the lack of effectiveness of daily messaging interventions. A bigger variation in demographics as well as a bigger sample size might be useful in future investigations into the idea that daily messaging appeals can promote PEB.

In this regard, it is important to keep in mind that people are free to choose whether to receive messages from climate advocacy groups in their daily life. It is likely that particularly those who decide to opt-in are already inclined to behave in an environmentally friendly way, whereas those most in need of intervention (i.e., people who are not environmentally friendly) avoid receiving and reading persuasive environmental appeals. Hence, it is not only important to address the effectiveness of environmental appeals but also to consider how people can be persuaded to opt-in to receiving appeals in the first place.

Previous daily messaging interventions used 14 messages and yielded significant effects (e.g., [10,11]), whereas the present intervention consisted of five messages only. Future studies

on how to promote PEB would be welcome to test the effectiveness of intervention with a longer duration. Another limitation is that the actual distinction between egoistic and ecological messages may have been less than intended (although a pilot study validated the messages in the present study), as even egoistical pro-environmental messages are of an ecological manner to some extent. Lastly, some of the present findings are based on participants' self-reported PEB, and self-reported PEB often differs from objective measures [19]. Future studies may examine the impact of egoistic and egocentric messages on objective indicators of people's PEB, such as the water consumption of households [30].

## 5. Conclusions

In sum, this research does not provide support for the effectiveness of a daily messaging technique to promote PEB. Although there was a tendency for ecological appeals in particular to enhance PEB, this effect was not statistically different from the increase that could also be observed in the other two conditions. Overall, it appears that 'being observed' was the most effective 'intervention'. In sum, at this point we have to conclude that it is less of a question of 'Ego or Eco' and more of people's awareness of being observed.

**Author Contributions:** Conceptualization, J.S.K. and T.G.; methodology, J.S.K.; formal analysis, J.S.K.; data curation, J.S.K.; writing—original draft preparation, J.S.K.; writing—review and editing, T.G.; All authors have read and agreed to the published version of the manuscript.

**Funding:** This research received no external funding.

**Conflicts of Interest:** The authors declare no conflict of interest.

## Appendix A

**Table A1.** Interventional messages, means (*M*) of egoistic and ecological appeal, *p*-values (Wilcoxon rank tests). Messages were originally presented in the German language.

| Message | $M_{ego}$ | $M_{eco}$ | $p$ |
|---|---|---|---|
| It is getting hot! In the winter of 2016 there were a number of days when it was 15 to 21 degrees Celsius warmer than normal. As a result, the North Pole was warmed. This also thawed the permafrost that surrounds the seed vault in Svalbard, Norway. Seeds from all over the world are stored in this safe. Nicknamed the 'Last Judgment', it is designed to ensure that the planet's agriculture could survive any disaster. Due to the partial thawing of the frozen ground, water ran into the seed stores—only ten years after it was set up, it was threatened by climate change' (Wallache-Wells, 2017). Think about it: how big do you estimate the collective loss would be if the seed vault was destroyed by climate change? | 3.71 | 6.00 | <0.001 |
| The proportion of carbon dioxide in the air will increase due to climate change. One consequence is as follows: The oceans are acidifying: about one quarter of $CO_2$ emitted is absorbed by the oceans, and their pH value is changed as a result. So far, this has already reduced the pH value of the oceans by 0.1. If the acidity of the oceans changes, this affects the ability of crustaceans to form shells or of corals to form reefs (see WMO, 2019, State of the Climate 2018). The sea is acidifying. The conditions are becoming more hostile to life (Global2000—https://www.global2000.at/haben-des-klimawandels). Imagine the seas without shellfish and coral. How will this influence the ecosystem? | 3.71 | 6.64 | <0.001 |
| Harmful insects can become a nuisance due to climate change. 'If you include harm of the bark beetle in the assessment of climate change effects on the spruce, there is a clear increase in very poorly suited forest areas. This is because the bark beetle favors a warmer climate, while at the same time, the spruce suffers from climatic stress and is therefore more prone to damage (WWF Klimastudie der Bundesforste AT, https://www.bundesforste.at/fileadmin/publikationen/studien/Klimastudie_WWF.pdf) | 4.11 | 6.32 | <0.001 |

**Table A1.** *Cont.*

| Message | $M_{ego}$ | $M_{eco}$ | $p$ |
|---|---|---|---|
| Melting ice: all over the world, ice is melting on the poles and glaciers. This is most visible in the Arctic sea ice. Expansion at the summer minimum in September is about 28% lower than average. The inland glaciers in Greenland are also melting. Since 2002, they have lost 3.600 billion tons of ice. In the last 10 years, the Swiss glaciers have lost half of their volume (see WMO (2019): Declaration on the state of the global climate). (Global2000—https://www.global2000.at/haben-des-klimawandels). Think about the ecological consequences of the disappearance of glaciers. | 4.18 | 6.64 | <0.001 |
| The ice is melting! Every ticket for flights from New York to London and back costs additional three-square meters of Arctic ice' (Wallache-Wells, 2017; according to Vizcarra. Stroeve & Notz, 2016). One consequence of this is that 'climate change could cause polar bears to become extinct by 2100. Due to rising temperatures, animals will find less food (Zdfheute/nature climate change, July 2020). Imagine this influence on the ecosystem and the consequences for polar bear. | 4.35 | 6.57 | <0.001 |
| Sometimes, just do not buy anything. Why do minimalists have less money worries? They consume less. They who spend little, save a lot. "Thrift is a good income", said Cicero over 2000 years ago. Owning things usually entails operating and maintenance costs. Disposal also has to be paid for at some point. That adds up in the end and is usually not considered when buying. (https://myfoodmyfuture.com/minimalismus/#Geld_sparen). Less is more—for your wallet! | 6.25 | 4.26 | <0.001 |
| Water is our most important food. It is obvious that we need to spend money on it. However, watch out: many people spend way too much money on their daily water. The cheapest mineral waters cost just under 15 cents per liter. Branded mineral waters are already around 70 cents. The scale is open at the top. Designer waters such as Voss sometimes cost 1.80 euros for half a liter. A liter of tap water costs less than 0.2 cents. People who stop buying water in plastic bottles do not only stop having to lug boxes and return bottles (and are, by the way, also kind to the environment), but they also save a lot of money (https://utopia.de/ratgeber/nachhaltiger-bio-konsum-mit-wenig-geld/). If you only replace buying 1 L of drinking water per day (70 cents) by drinking tap water (0.2 cents), you save around 19 euros a month. That is a lot of money every year! | 6.21 | 4.64 | <0.001 |
| The ice is melting! There are diseases in the Arctic ice that have not circulated in our air for millions of years—some of them were already around before humans existed. That means that our immune system would have no idea how it should ward off these prehistoric diseases, if they should be released again. In the Arctic, however, there are also terrifying germs from more recent times. In Alaska, researchers have found remains of the 1918 flu, which infected 500 million people and killed up to 100 million people—that's five percent of the world's population at the time and almost six times as many as were killed in the First World War (...). In May the BBC reported that scientists also suspected the presence of smallpox and the bubonic plague in Siberian ice' (Wallache-Wells, 2017). Imagine what restrictions the release of the diseases in the Arctic ice could place on your daily life. | 6.00 | 5.14 | <0.05 |
| Walking, taking the stairs instead of the elevator and cycling to work are not only good for the environment but primary pays off for you personally. 'The advantages are: prevention of back pain and cardiovascular diseases, diabetes, arthrosis, osteoporosis, depression and others; prevention of obesity or helping you lose weight; increases self-confidence and well-being; improves performance; promotes good sleep; saves money because cars are no longer needed (Https://myfoodmyfuture.com/minimalismus/#Tipps_Minimalismus_entruempeln). | 6.64 | 3.96 | <0.001 |

**Table A1.** *Cont.*

| Message | $M_{ego}$ | $M_{eco}$ | $p$ |
|---|---|---|---|
| The proportion of carbon dioxide in the air will increase due to climate change. One consequence is Air that you can't breathe—our lungs need oxygen. However, this is only a fraction of what we breathe. For example, the proportion of carbon dioxide in the air is increasing. It has just risen above 400 parts per million (ppm). Estimates based on current trends suggest that it will be at 1000 ppm by the end of the century. That concentration would, compared with the air that we breathe today, lead to a 21 percent decline in people's cognitive abilities (Wallache-Wells, 2017). Think about how air that you can't breathe would feel. | 5.96 | 5.54 | =0.09 |

## Appendix B

**Table A2.** Questionnaire on pro-environmental behavior, items and correlation to the full scale's score (Cronbach's $\alpha_{pre}$ = 0.77 [0.72, 0.81]; Cronbach's $\alpha_{post}$ = 0.81 [0.76, 0.84]; $N$ = 206).

| "In the Last 5 days, I have … " (Items) | $r_{pre}$ | $r_{post}$ |
|---|---|---|
| used detergents in a smaller dose, than manufacturers recommended (e.g., while dishwashing, cleaning, … ). | 0.59 | 0.64 |
| saved water in my household for primary environmental reasons (e.g., short showering instead of taking a bath). | 0.58 | 0.58 |
| taken recyclable materials to the adequate recycling stations (e.g., paper, glass, aluminum). | 0.40 | 0.39 |
| forgone eating meat, for environmental reasons. | 0.52 | 0.54 |
| preferred regionally grown food for primary environmental reasons. | 0.62 | 0.72 |
| mainly chosen environmentally friendly transportation alternatives (e.g., train or bicycle instead of car). | 0.43 | 0.39 |
| chosen stairs over elevators to save energy for environmental reasons. | 0.51 | 0.58 |
| taken a bag with me for shopping to avoid taking a plastic bag in shop. | 0.43 | 0.48 |
| avoided plastic that is used only once while shopping (e.g., food packages). | 0.60 | 0.63 |
| avoided buying drinks in cans or plastic bottles for environmental reasons. | 0.67 | 0.65 |
| payed attention to labels for ecological products (on food, clothes, … ). | 0.62 | 0.69 |
| turned off lights and power switches when they are not needed to avoid unnecessary energy use. | 0.43 | 0.38 |

*Note.* Adapted to the English language, originally presented in German.

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
