# Peer review of "Ego or Eco? Neither Ecological nor Egoistic Appeals of Persuasive Climate Change Messages Impacted Pro-Environmental Behavior"

_sustainability, doi:10.3390/su122310064_

Round 1
Reviewer 1 Report
The study sought to test whether egoistic or ecological appeals had greater impacts on increasing PEB. This is a worthwhile pursuit in the effort of improving messaging to address climate change. The research design made an interesting use of text messaging to relay the messages to participants. Ultimately, no differences were found in the types of messages, and rather, the knowledge of being watched seemed to have the greatest impact on increasing PEBs.
The paper is generally well written and addresses an important issue around climate messaging. I support publishing null results, in cases where the research design is strong enough to infer that the null effects were a result of an actual lack of importance rather than artifacts of the design. The paper leaves out several areas of the study's design that I would like to see before being able to judge the usefulness of the findings.
- Paper states that participants were recruited in three ways using advertisements. I would like to see the message potential participants saw asking for participation. Did it convey that the study was about environmental behaviors? That seems like it would make it more likely to recruit people with an interest in environmentalism and may contribute to the lack of effect because of a strong floor.
- At least in the appendix I would like to see the attention check items. The messages struck me as being rather long for receiving as a text and I wonder how carefully they would actually be read. Knowing that there was a strong attention check would relieve some of these concerns.
- I appreciated the validation of the intervention messages with pilot studies. However, the distinction between ego and eco was not clear; what I mean is that they didn't really seem like an either/or message. The ego message, because it makes reference to climate impacts, also primes eco concerns. And, probably to a lesser extent, the eco message may prime egoistic concerns about life on a changed planet. The pilot studies show this continuum in how respondents rated the messages. The point is, the null effects could be because of this lack of a clear distinction between ego and eco, not because the actual distinction between those concepts doesn't matter. The paper should more fully address this possibilty.
Some other concerns:
- The authors state that they performed a power analysis to determine sample size. I understood that was for detecting the main effect. If so, then the interaction analyses would be considerably under-powered. Gelman (2018) recommends 16x the sample size for an interaction.
- I wonder about the external validity of the study. Climate advocacy groups or even governmental entities may wish to use this type of messaging to increase PEBs. Would daily text messages actually be realistic? If people opted-in, whether out of interest or because they are members of a group, wouldn't they more likely to engage in PEBs anyway? The authors should address this in the conclusion.
- Self-reported PEB has limitations, but I accept it as valid, especially in larger N studies. I would like to see more discussion of the measurement of PEB. Were the items adopted from another article? Beyond Cronbach's alpha were they validated in any other way (eg: pilot testing, factor analysis)? I worry that mentioning 'environment' within many of the items contributes to social desirability bias and/or that it would make DV endogenous to environmental attitude.
- Why was PEB higher for the control group? That finding makes me wonder about the variance with a smallish sample size.
A final minor point on writing. I prefer using male/female as adjectives rather than nouns and recommend referring to men and women, rather than males and females.
References:
Gelman, 2018: https://statmodeling.stat.columbia.edu/2018/03/15/need-16-times-sample-size-estimate-interaction-estimate-main-effect/
Author Response
We very much appreciate the feedback the reviewers have provided and have revised the manuscript accordingly. In particular, we have changed and restructured parts of the introduction and the discussion (see our detailed responses to the reviewers’ comments).
Below we provide the revision report, along with our responses to each of the points that were raised; our responses are marked with an asterisk. Please feel free to contact us if you have any questions about this resubmission.
The study sought to test whether egoistic or ecological appeals had greater impacts on increasing PEB. This is a worthwhile pursuit in the effort of improving messaging to address climate change. The research design made an interesting use of text messaging to relay the messages to participants. Ultimately, no differences were found in the types of messages, and rather, the knowledge of being watched seemed to have the greatest impact on increasing PEBs.
The paper is generally well written and addresses an important issue around climate messaging. I support publishing null results, in cases where the research design is strong enough to infer that the null effects were a result of an actual lack of importance rather than artifacts of the design. The paper leaves out several areas of the study's design that I would like to see before being able to judge the usefulness of the findings.
* Thank you for your very helpful advice. We added information about the study’s design (see the following points).
Paper states that participants were recruited in three ways using advertisements. I would like to see the message potential participants saw asking for participation. Did it convey that the study was about environmental behaviors? That seems like it would make it more likely to recruit people with an interest in environmentalism and may contribute to the lack of effect because of a strong floor.
* We agree with the reviewer that the message to invite potential participants is crucial to avoid selection biases. We added the following information (line 168-70: “The invitation stated it was a study about “personality and behavior in everyday life”. To minimize selection biases, the invitation did not say that the study was about environmental attitudes or behavior.”
At least in the appendix I would like to see the attention check items. The messages struck me as being rather long for receiving as a text and I wonder how carefully they would actually be read. Knowing that there was a strong attention check would relieve some of these concerns.
* We completely agree and thus further explained the attention check in lines 176-178: “They were asked if specific contents (e.g., melting of arctic ice [true], electrified motorization [false]) were part of the messages they received. If more than two out of seven answers were wrong, the participant’s data were excluded from further analyses.”
I appreciated the validation of the intervention messages with pilot studies. However, the distinction between ego and eco was not clear; what I mean is that they didn’t really seem like an either/or message. The ego message, because it makes reference to climate impacts, also primes eco concerns. And, probably to a lesser extent, the eco message may prime egoistic concerns about life on a changed planet. The pilot studies show this continuum in how respondents rated the messages. The point is, the null effects could be because of this lack of a clear distinction between ego and eco, not because the actual distinction between those concepts doesn’t matter. The paper should more fully address this possibility.
* Thank you very much for pointing this out. We now address this possibility in the discussion, lines 351-353: “Another limitation is that the actual distinction between egoistic and ecological messages may have been less than intended (although a pilot study validated the messages in the present study), as even egoistical pro-environmental messages are of ecological manner to some extent.”
Some other concerns:
The authors state that they performed a power analysis to determine sample size. I understood that was for detecting the main effect. If so, then the interaction analyses would be considerably under-powered. Gelman (2018) recommends 16x the sample size for an interaction.
* Thank you for this comment and the advised reference. As we conducted a repeated measurement ANOVA, the Gelman recommendation is not applicable for our design. We added information about the underlying power analysis in line 141-146: “A G*Power Analysis [16] (MANOVA for a two times repeated measurement in three groups including interaction effects, 95% Power) revealed a sample size of 198 participants for an expected medium effect size of f = 0.20, based on previous findings from Carfora et al. [10, 11] who found that both a health and an environment daily messaging condition reduced red and processed meat consumption compared to a control condition (d = 0.53, d = 0.47, respectively).”
I wonder about the external validity of the study. Climate advocacy groups or even governmental entities may wish to use this type of messaging to increase PEBs. Would daily text messages actually be realistic? If people opted-in, whether out of interest or because they are members of a group, wouldn't they more likely to engage in PEBs anyway? The authors should address this in the conclusion.
* We appreciate this recommendation and thus added a part of the discussion (lines 334-340: “First, the sample primary consisted of young and highly educated women, which could explain why values of PEB were already high from the beginning, as it has been shown that women more than men behave in an environmentally-friendly way [26] and that education had a positive impact on PEB [27]. This ceiling effect, in turn, may account for the lack of effectiveness of the daily messaging interventions. A bigger variation in demographics as well as a bigger sample size might be useful in future replications of whether daily messaging appeals can promote PEB.”
We also noted that it is questionable that those most in need of intervention (i.e., people who are not environmentally friendly) avoid receiving and reading persuasive environmental appeals and that it should be also considered how people can be persuaded to opt-in receiving those appeals in the first place (lines 341-347).
Also, we now address the practical application of the present study: “We are all target of a daily message-flood. Among the information reaching us in daily life through all sorts of media and adverts, there are many messages relating to climate crisis and sustainability [e.g., 4]. Therefore, it is important to investigate which, and if, such information can trigger PEB in daily life.” (lines 37-40)
Self-reported PEB has limitations, but I accept it as valid, especially in larger N studies. I would like to see more discussion of the measurement of PEB. Were the items adopted from another article? Beyond Cronbach's alpha were they validated in any other way (eg: pilot testing, factor analysis)? I worry that mentioning 'environment' within many of the items contributes to social desirability bias and/or that it would make DV endogenous to environmental attitude.
* We now describe the scale that was used to measure PEB in more detail, line 162-165: “PEB of participants was measured by a questionnaire before (pre) and after (post) the intervention (all items and information about reliability can be found in appendix A). The questionnaire was based on previous validated scales [18, 19] and extended by self-developed items. All items targeted behavior that was shown for primary environmental reasons.”
Why was PEB higher for the control group? That finding makes me wonder about the variance with a smallish sample size.
* Indeed, we assume that this is a matter of coincidence. Accordingly, we now describe differences between groups and how we addressed this issue (line 219-225): “Groups did not differ in terms of age (F(2, 203) = 1.00, p = .370, η2 = .01), gender ((F(2, 203) = 0.33, p = .721, η2 = .00), education (F(2, 203) = 0.65, p = .526, η2 = .01), political attitude (F(2, 203) = 2.28, p = .105, η2 = .02) or pro-environmental attitude (NEP mean; F(2, 203) = 0.81, p = .445, η2 = .01). In contrast, the experimental groups differed in PEB at time one, F(2, 203) = 3.39, p = .036, η2 = .03. The control group reported significantly higher values than the ego-group (p = .034) and the eco group (p = .024). At time two, this difference disappeared, F(2, 203) = 1.92, p = .150, η2 = .02.”
A final minor point on writing. I prefer using male/female as adjectives rather than nouns and recommend referring to men and women, rather than males and females.
* We strongly agree with the reviewer and accordingly reworded these phrases (line 149 and 336).
References:
Gelman, 2018: https://statmodeling.stat.columbia.edu/2018/03/15/need-16-times-sample-size-estimate-interaction-estimate-main-effect/

Reviewer 2 Report
The manuscript addresses the timely topic of promoting sustainable behaviour. The study is well-planned and a strength is that it was preregistered and the necessary sample size to detect a medium effect was determined a priori. However, there are several issues regarding the reporting that can be improved. Please find my comments below.
- Introduction: The structure of the introduction could be improved. Currently, the authors first summarise the literature briefly before then citing further studies on persuasive messages. The lack of structure is also indicated by a need to refer to a later paragraph in the text in line 43. Furthermore, there is only one subheading (1.1), which should be avoided.
- Introduction: Moreover, the introduction could be condensed. Currently the section 1.1 describes several studies at length. Such an extensive summary is not needed to follow the main argument of the introduction.
- Introduction: In RQ1, the authors suddenly mention social value orientation, which was not introduced before. A paragraph on social value orientation should be added before the hypothesis and research questions are listed.
- Methods: Please provide information about ethical approval, informed consent and compensation.
- Methods: Did demographics differ between groups?
- What were the poles/ anchors on the political orientation scale?
- Methods: Is subheading 2.2 supposed to read “Design and Measures”? Having a subheading Methods in the methods section seems unnecessary.
- Methods: A section on data analysis is missing.
- Methods: Social value orientation is missing from Table 1.
- Results: It would be helpful to repeat H1 before reporting the analysis.
- Results: For H1 the authors should highlight that they analysed simple main effects based on the significant main effect for time, otherwise the reporting may be confusing given the non-significant interaction. The abstract should be adapted accordingly, stating first that there was a no significant interaction between messaging condition and time, but a significant main effect for time that was followed up by simple main effects.
- Results: Recoding of variables should be described in the methods section, either in a subsection on measures or in a subsection on data analysis.
- Results. How was “primary prosocial/ individualistic value orientation” determined based on a continuous scale?
- Results: Dummy coding of variables is usually 0/1 to allow for easier interpretation of the values.
- Figure 1: The y-axis is misleading – it should start at the minimum possible value.
- Discussions: The authors might want to refer to this effect as question-behavior-effect (instead of Hawthorne effect), given that the cited systematic review by McCambridge et al. shows that the label Hawthorne effect has been used for a variety of manipulations. The following two systematic reviews on the question-behavior effect might be of interest:
Miles, L. M., Rodrigues, A. M., Sniehotta, F. F., & French, D. P. (2020). Asking questions changes health-related behaviour: An updated systematic review and meta-analysis. Journal of Clinical Epidemiology.
Wilding, S., Conner, M., Sandberg, T., Prestwich, A., Lawton, R., Wood, C., ... & Sheeran, P. (2016). The question-behaviour effect: a theoretical and methodological review and meta-analysis. European Review of Social Psychology, 27(1), 196-230.
- Discussion: Values for PEB were already quite high (means between 5.6 and 6.15 for a scale with a maximum value of 7). How do these values compare to other studies, especially given the sample characteristics? Could the lack of effectiveness be due to a ceiling effect?
- Preregistration: The reported study diverges from the preregistration in several aspects: willingness to donate was not reported; unsure if ANOVA or MANOVA was used; sample size calculation (different effect sizes and test reported). Please highlight the divergences and explain why you chose to diverge from the protocol.
Author Response
We very much appreciate the feedback the reviewers have provided and have revised the manuscript accordingly. In particular, we have changed and restructured parts of the introduction and the discussion (see our detailed responses to the reviewers’ comments).
Below we provide the revision report, along with our responses to each of the points that were raised; our responses are marked with an asterisk. Please feel free to contact us if you have any questions about this resubmission.
Reviewer #2
The manuscript addresses the timely topic of promoting sustainable behaviour. The study is well-planned and a strength is that it was preregistered and the necessary sample size to detect a medium effect was determined a priori. However, there are several issues regarding the reporting that can be improved. Please find my comments below.
* Thank you for your very helpful recommendations.
Introduction: The structure of the introduction could be improved. Currently, the authors first summarise the literature briefly before then citing further studies on persuasive messages. The lack of structure is also indicated by a need to refer to a later paragraph in the text in line 43. Furthermore, there is only one subheading (1.1), which should be avoided.
* We strongly agree with the reviewer and thus restructured the introduction. By adding information about moderator variables (see below) and cutting down the paragraph about previous studies (see below), we also added subheadings (lines 56, 84, 107).
Introduction: Moreover, the introduction could be condensed. Currently the section 1.1 describes several studies at length. Such an extensive summary is not needed to follow the main argument of the introduction.
* We appreciate this recommendation and cut down the section “Previous research on the use of persuasive messages to promote PEB” (lines 59-86).
Introduction: In RQ1, the authors suddenly mention social value orientation, which was not introduced before. A paragraph on social value orientation should be added before the hypothesis and research questions are listed.
* As already mentioned, we added a section about “1.2 Potential moderator variables” (line 87) whereas not only social value orientation, but also the other potential moderator variables are now described in more detail.
Methods: Please provide information about ethical approval, informed consent and compensation.
* We agree with the reviewer and added information about consent and compensation (lines 196-198): “At the end of the study, participants could take part in a raffle (five times 50 Euro) as a reward and register for a detailed briefing of the study aims and results.” Ethical approval and funding information is now given at the end of the manuscript (lines 471-472).
Methods: Did demographics differ between groups?
* We now describe differences between groups (line 219-225): “Groups did not differ in terms of age (F(2, 203) = 1.00, p = .370, η2 = .01), gender ((F(2, 203) = 0.33, p = .721, η2 = .00), education (F(2, 203) = 0.65, p = .526, η2 = .01), political attitude (F(2, 203) = 2.28, p = .105, η2 = .02) or pro-environmental attitude (NEP mean; F(2, 203) = 0.81, p = .445, η2 = .01). In contrast, the experimental groups differed in PEB at time one, F(2, 203) = 3.39, p = .036, η2 = .03. The control group reported significantly higher values than the ego-group (p = .034) and the eco group (p = .024). At time two, this difference disappeared, F(2, 203) = 1.92, p = .150, η2 = .02.”
What were the poles/ anchors on the political orientation scale?
* We added this information in lines 155-156 (Mean political orientation was 4.06 (SD = 1.63) given on a slider from“1 - politically left” to “11 – politically right”.)
Methods: Is subheading 2.2 supposed to read “Design and Measures”? Having a subheading Methods in the methods section seems unnecessary.
* We strongly agree and restructured the method section by the following sub-headings: 2.1 Participants, 2.2 Design, 2.3 Materials and 2.4 Data Analysis
Methods: A section on data analysis is missing.
* see comment above
Methods: Social value orientation is missing from Table 1.
* As there were no single items and no scale reliability that can be reported, we still abstained from reporting the social value orientation measure in the table, but rather provided more information about this measure in the main text (lines 200-204).
Results: It would be helpful to repeat H1 before reporting the analysis.
* We now mention H1 in the result section (line 233 “To test whether interventional messages enhance PEB (…)”)
Results: For H1 the authors should highlight that they analysed simple main effects based on the significant main effect for time, otherwise the reporting may be confusing given the non-significant interaction. The abstract should be adapted accordingly, stating first that there was a no significant interaction between messaging condition and time, but a significant main effect for time that was followed up by simple main effects.
* We agree with the reviewer. Accordingly, line 240 is now reworded. In addition, we restructured this sequence in the abstract (lines 17-23).
Results: Recoding of variables should be described in the methods section, either in a subsection on measures or in a subsection on data analysis.
* As already mentioned, we added the sub-heading “2.4 Data Analysis”, where the coding of variables is now described (lines 200-208).
Results. How was “primary prosocial/ individualistic value orientation” determined based on a continuous scale?
* We added the following information (lines 200-208): “Regarding the participant’s social value orientation, a decomposed game [13] included nine decisions about a distribution of points to either oneself or another person. The predominant type of decisions was coded as the individual’s social value orientation. In total, there were 37 participants with a proself orientation, 157 with a prosocial orientation, eight with a competitive orientation and sixteen participants with an inconclusive orientation. Further, we coded fit by “1” (N = 63) when participants with a primary prosocial value orientation received ecological appeals and when participants with a primary individualistic value orientation received egoistic appeals, respectively. Non-fit was coded with zero (N = 61).”
Results: Dummy coding of variables is usually 0/1 to allow for easier interpretation of the values.
* We now use 0/1 for coding of variables (lines 204-208).
Figure 1: The y-axis is misleading – it should start at the minimum possible value.
* The y-axis now starts at the minimum possible value (Figure 1).
Discussions: The authors might want to refer to this effect as question-behavior-effect (instead of Hawthorne effect), given that the cited systematic review by McCambridge et al. shows that the label Hawthorne effect has been used for a variety of manipulations. The following two systematic reviews on the question-behavior effect might be of interest:
Miles, L. M., Rodrigues, A. M., Sniehotta, F. F., & French, D. P. (2020). Asking questions changes health-related behaviour: An updated systematic review and meta-analysis. Journal of Clinical Epidemiology.
Wilding, S., Conner, M., Sandberg, T., Prestwich, A., Lawton, R., Wood, C., ... & Sheeran, P. (2016). The question-behaviour effect: a theoretical and methodological review and meta-analysis. European Review of Social Psychology, 27(1), 196-230.
* Thank you very much for this very helpful comment and the given references. We now refer to the question-behavior-effect instead of the Hawthorn effect, lines 280-283: “It thus appears that the main finding of our study is that study participation itself enhanced PEB. Such research participation effects [21] have been coined the “question-behavior effect” [22, 23], describing the pattern that participants change their behavior when questions are being asked about it.”
Discussion: Values for PEB were already quite high (means between 5.6 and 6.15 for a scale with a maximum value of 7). How do these values compare to other studies, especially given the sample characteristics? Could the lack of effectiveness be due to a ceiling effect?
* We now address this issue as a part of the discussion, lines 334-347: “First, the sample primary consisted of young and highly educated women, which could explain why values of PEB were already high from the beginning, as it has been shown that women more than men behave in an environmentally friendly way [26] and that education had a positive impact on PEB [27]. This ceiling effect, in turn, may account for the lack of effectiveness of the daily messaging interventions. A bigger variation in demographics as well as a bigger sample size might be useful in future investigations of the idea that daily messaging appeals can promote PEB.
In this regard, it is important to keep in mind that people in their daily life are free to choose whether to receive messages from climate advocacy groups. It is likely that particularly those decide to opt-in who are already inclined to behave in an environmentally friendly way, whereas those most in need of intervention (i.e., people who are not environmentally friendly) avoid receiving and reading persuasive environmental appeals. Hence, it is not only important to address the effectiveness of environmental appeals, but also to consider how people can be persuaded to opt-in receiving those appeals in the first place.”
Preregistration: The reported study diverges from the preregistration in several aspects: willingness to donate was not reported; unsure if ANOVA or MANOVA was used; sample size calculation (different effect sizes and test reported). Please highlight the divergences and explain why you chose to diverge from the protocol.
* We appreciate this comment and now highlight differences in a footnote (p. 3): “As another dependent variable (noted in the pre-registration), donation behavior of a potential prize money was assessed. However, because no significant effects were found, this variable is not considered further. Before any analyses were performed, we calculated another power analysis (the expected effect size was lowered from f = .23 to f = .20) and thus enhanced sample size.”
Power analysis is described in lines 141-146: “A G*Power Analysis [17] (MANOVA for a two times repeated measurement in three groups including interaction effects, 95% Power) revealed a sample size of 198 participants for an expected medium effect size of f = 0.20, based on previous findings from Carfora et al. [11, 12] who found that both a health and an environment daily messaging condition reduced red and processed meat consumption compared to a control condition (d = 0.53, d = 0.47, respectively).”

Round 2
Reviewer 1 Report
I commend the authors for thoroughly addressing the concerns raised in my first review and am mostly satisfied with the current state of the paper. I do have a few minor suggestions, mostly regarding the clarity of the writing.
Line 118: Looks like a minor formatting error. I’m guessing “1.3 Hypothesis” should be on the next line?
Line 155: Missing a space between from, and “. It also may just be the font or how it is displaying on my computer, but it looks like the front quotation marks are going in the wrong direction. Please double-check.
Line 163-4: Sentence would read more clearly as: “The questionnaire was based on previously validated scales [18, 19] and some newly developed items for this study.
Line 164-5: The last sentence of the paragraph is awkwardly phrased. I suggest, “All items targeted behaviors that are undertaken primarily for environmental reasons.” And include a citation.
Line 169: Period should be placed inside quotation marks.
Line 171: The sentence “PEB was assessed retrospectively for the last five days” is redundant with 174. Delete.
Line 194: The current NEP refers to the New Ecological Paradigm.
First paragraph of the data analysis section is very helpful.
Line 245-246: I suggest a change in wording: “Overall, however, H1 was not supported by the data.”
Discussion: I agree with the other reviewer on the recommendation to use the more specific “question-behavior effect” terminology.
Author Response
We very much appreciate the feedback the reviewers have provided once more.
Below we provide the revision report (round two), along with our responses to each of the points that were raised; our responses are marked with an asterisk.
A new version of the manuscript is attached.
----
I commend the authors for thoroughly addressing the concerns raised in my first review and am mostly satisfied with the current state of the paper. I do have a few minor suggestions, mostly regarding the clarity of the writing.
* We very much appreciate your suggestions and thoroughly addressed them (see below).
Line 118: Looks like a minor formatting error. I’m guessing “1.3 Hypothesis” should be on the next line?
* Thank you for pointing this out. We fixed that (line 134, now renamed due to the recommendation of another reviewer).
Line 155: Missing a space between from, and “. It also may just be the font or how it is displaying on my computer, but it looks like the front quotation marks are going in the wrong direction. Please double-check.
* We corrected the quotation marks and added a space.
Line 163-4: Sentence would read more clearly as: “The questionnaire was based on previously validated scales [18, 19] and some newly developed items for this study.
* Thank you for your recommendation. We accordingly adopted this sentence (lines 181-182).
Line 164-5: The last sentence of the paragraph is awkwardly phrased. I suggest, “All items targeted behaviors that are undertaken primarily for environmental reasons.” And include a citation.
* We have adopted the proposed formulation. Citation now links to the exact formulations attached in the appendix A (lines 182-183).
Line 169: Period should be placed inside quotation marks.
* We added quotation marks.
Line 171: The sentence “PEB was assessed retrospectively for the last five days” is redundant with 174. Delete.
* Thank you for this comment, we deleted this sentence.
Line 194: The current NEP refers to the New Ecological Paradigm.
* We fixed that (line 212).
First paragraph of the data analysis section is very helpful.
* Thank you.
Line 245-246: I suggest a change in wording: “Overall, however, H1 was not supported by the data.”
* Thank you for your recommendation. We adopted the proposed formulation (lines 275-276).
Discussion: I agree with the other reviewer on the recommendation to use the more specific “question-behavior effect” terminology.

Reviewer 2 Report
The revisions have significantly improved the manuscript. A few remaining issues are listed below.
- line 118: line break is missing; given that there were both concrete hypotheses and undirected research questions I would suggest to label this subsection “The present study”.
- Results of pre-registered test should be reported irrespective of the significance of the findings. Especially reporting non-significant findings are important to alleviate the file drawer problem. Please report the rationale for including donation behaviour and the non-significant findings.
- lines 131ff: The square brackets should be omitted.
- The game to determine social value orientation should be reported together with the other measures and not under data analysis.
Author Response
We very much appreciate the feedback the reviewers have provided once more.
Below we provide the revision report (round two), along with our responses to each of the points that were raised; our responses are marked with an asterisk.
A new version of the manuscript is attached.
----
The revisions have significantly improved the manuscript. A few remaining issues are listed below.
- line 118: line break is missing; given that there were both concrete hypotheses and undirected research questions I would suggest to label this subsection “The present study”.
* We followed the reviewer’s advice.
- Results of pre-registered test should be reported irrespective of the significance of the findings. Especially reporting non-significant findings are important to alleviate the file drawer problem. Please report the rationale for including donation behaviour and the non-significant findings.
* We added a description of the additional dependent variable (willingness to donate), line 220-226: “At the end of the study, participants could take part in a raffle (five times 50 Euro) as a reward and register for a detailed briefing of the study aims and results. We assessed the participant’s willingness to donate parts of the potential prize money to prosocial organizations (“NEUstart”, the Austrian probation service or “Austrian Cancer Aid”) and/or pro-environmental organizations (“Protect our Winters Austria” or “Fridays for Future Austria”). By the end of the study, five winners were drawn and the prize money was paid out to themselves and the organizations according to the division they stated earlier.”
In addition, findings for the willingness to donate to prosocial and pro-environmental organizations variable are added in Table 2, as well as in Figure 2. Results are reported in lines 272-276, 285-287 and 301-314. We also made changes to the abstract (lines 15-18).
- lines 131ff: The square brackets should be omitted.
* We fixed that.
- The game to determine social value orientation should be reported together with the other measures and not under data analysis.
* Thank you for this recommendation. We moved this paragraph to lines 214-219, under the method/materials section.

Round 3
Reviewer 2 Report
By including all tested hypotheses the manuscript wasa further improved. I have no further comments.
Author Response
Thank you again, for your very helpful comments.
